# SHUFFLE TO LEARN: SELF-SUPERVISED LEARNING FROM PERMUTATIONS VIA DIFFERENTIABLE RANKING

## ABSTRACT

Self-supervised pre-training using so-called "pretext" tasks has recently shown impressive performance across a wide range of tasks. In this work, we advance self-supervised learning from permutations, that consists in shuffling parts of input and training a model to reorder them, improving downstream performance in classification. To do so, we overcome the main challenges of integrating permutation inversions (a discontinuous operation) into an end-to-end training scheme, heretofore sidestepped by casting the reordering task as classification, fundamentally reducing the space of permutations that can be exploited. We make two main contributions. First, we use recent advances in differentiable ranking to integrate the permutation inversion flawlessly into a neural network, enabling us to use the full set of permutations, at no additional computing cost. Our experiments validate that learning from all possible permutations improves the quality of the pre-trained representations over using a limited, fixed set. Second, we successfully demonstrate that inverting permutations is a meaningful pretext task in a diverse range of modalities, beyond images, which does not require modality-specific design. In particular, we improve music understanding by reordering spectrogram patches in the time-frequency space, as well as video classification by reordering frames along the time axis. We furthermore analyze the influence of the patches that we use (vertical, horizontal, 2-dimensional), as well as the benefit of our approach in different data regimes.

## 1 INTRODUCTION

Supervised learning has achieved important successes on large annotated datasets (Deng et al., 2009; Amodei et al., 2016). However, most available data, whether images, audio, or videos are unlabelled. For this reason, pre-training representations in an unsupervised way, with subsequent fine-tuning on labelled data, has become the standard to extend the performance of deep architectures to applications where annotations are scarce, such as understanding medical images (Rajpurkar et al., 2017), recognizing speech from under-resourced languages (Rivière et al., 2020; Conneau et al., 2020), or solving specific language inference tasks (Devlin et al., 2018). Among unsupervised training schemes, self-supervised learning focuses on designing a proxy training objective, that requires no annotation, such that the representations incidentally learned will generalize well to the task of interest, limiting the amount of labeled data needed for fine-tuning. Such "pretext" tasks, a term coined by Doersch et al. (2015), include learning to colorize an artificially gray-scaled image (Larsson et al., 2017), inpainting removed patches (Pathak et al., 2016) or recognizing with which angle an original image was rotated (Gidaris et al., 2018). Other approaches for self-supervision include classification to original images after data augmentation (Chen et al., 2020) and clustering (Caron et al., 2018).

In this work, we consider the pretext task of reordering patches of an input, first proposed for images by Noroozi & Favaro (2016), the analogue of solving a jigsaw puzzle. In this setting, we first split an input into patches and shuffle them by applying a random permutation. We train a neural network to predict which permutation was applied, taking the shuffled patches as inputs. We then use the inner representations learned by the neural network as input features to a low-capacity supervised classifier (see Figures 1 and 2 for illustration). We believe that permutations provide a promising avenue for self-supervised learning, as they are conceptually general enough to be applied across a large range of modalities, unlike colorization (Larsson et al., 2017) or rotations (Gidaris et al., 2018) that are specific to images. The idea of using permutations was also explored in Santa Cruz et al.

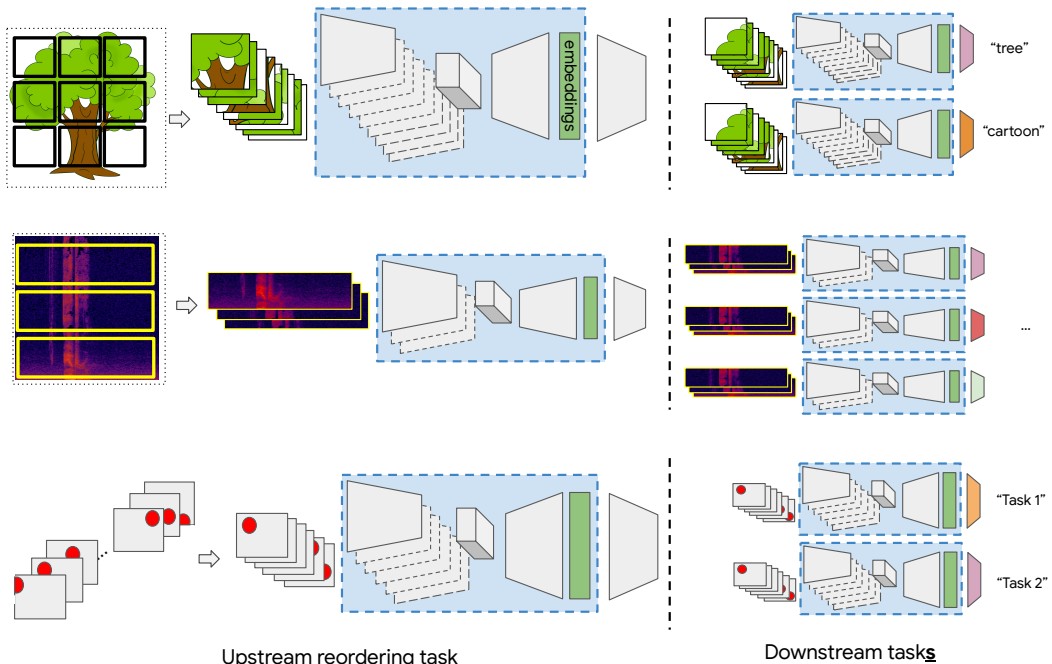

Figure 1: Permutations as a self-supervised technique can handle a variety of modalities with minimal changes to the network architecture. Dotted layers indicate weight sharing across input patches. The learned embeddings can be used for several downstream tasks. The inputs for the upstream task are permuted, while the inputs for the downstream task are not.

(2018) where they use a bi level optimization scheme which leverages sinkhorn iterations to learn visual reconstructions. Their method resorts to approximating the permuation matrix with such continuous methods. Our method relies on no such approximations and can efficiently represent all possible permutations. Moreover, the encouraging results of Noroozi & Favaro (2016) when transferring learned image features for object detection and image retrieval inspire us to advance this method a step forward. However, including permutations into an end-to-end differentiable pipeline is challenging, as permutations are a discontinuous operation. Noroozi & Favaro (2016) circumvent this issue by using a fixed set of permutations and casting the permutation prediction problem as a classification one. Given that the number of possible permutations of $n$ patches is $n!$, this approach cannot scale to exploiting the full set of permutations, even when $n$ is moderately small.

In this work, we leverage recent advances in differentiable ranking (Berthet et al., 2020; Blondel et al., 2020b) to integrate permutations into end-to-end neural training. This allows us to solve the permutation inversion task for the entire set of permutations, removing a bottleneck that was heretofore sidestepped in manners that could deteriorate downstream performance. Moreover, we successfully demonstrate for the first time the effectiveness of permutations as a pretext task on multiple modalities with minimal modality-specific adjustments. In particular, we improve music understanding by learning to reorder spectrogram frames, over the time and frequency axes. We also improve video understanding by reordering video frames along time.

To summarize, we make the following two contributions.

- We integrate **differentiable ranking** into end-to-end neural network training for representation learning. This provides an efficient manner to learn in reordering tasks for all permutations, for larger numbers of patches. We show that this drastic increase in the number of permutations improves the quality of learned representations for downstream tasks.

- We successfully demonstrate for the first time the effectiveness of permutations as a **general purpose self-supervision method**, efficient on multiple modalities with extremely minimal modifications to the network. Additionally, the pre-trained representations perform well across diverse

Figure 2: Data processing and upstream training procedures. All permutations are sampled on the fly, during training. To obtain embeddings reusable for downstream tasks, we truncate the network by removing the last few layers. This truncation depends on the task and is explained in Section 3.

tasks of the same modality. We purposefully divert from domain-specific transformations, often inspired by data augmentation techniques, as predicting the permutation applied to input patches is not restricted either to a domain, nor a modality or a dimensionality. This also creates opportunities for applications beyond the scope of what we illustrate in our work.

The rest of the paper is organized as follows. In Section 2 we present the problem formulation and the methods used. In Section 3 we demonstrate the effectiveness of our experiments on audio, video, and image tasks. Further details can be found in the Appendix.

## 2 METHODS

### 2.1 GENERAL METHODOLOGY

To efficiently leverage the existence of large amounts of unlabeled data, we present a self-supervised pretext task that predicts the permutation applied to patches of an input. We do so in a manner that removes an existing bottleneck, and allows to use all possible permutations as targets during training. This pretext task is performed *upstream* and the internal representation learned by the pretext neural network can then be transferred and used on secondary *downstream* tasks – see Figures 1 and 2.

In the upstream task, for each data point, $n$ *patches*, sub-parts $x_1, \ldots, x_n$ of of identical dimensions $d$ are extracted. Their exact structure depend naturally on the modality: e.g. horizontal bands of an image, frequency bands of a spectrogram, frames of a video. Accordingly, the dimensions in $d$ can represent height, width, channels, etc. These patches are then permuted randomly, and organized in a tensor $X_i$ of dimension $n \times d$ (see Figure 2), which is paired to the applied permutation as a label $y_i$ (see Section 2.2 for details on permutation encoding).

We then train the weights $w$ of a neural network to minimize the loss between its output $f_w(X_i)$, of size $n$, and the encoding $y_i \in \mathbb{R}^n$ of the permutation applied to the $n$ patches (see Figure 2). Note that the last operation of the network is a differentiable ranking operator $y_\varepsilon^*$. This operator, the encoding of the permutations for the labels, and the loss used in this upstream task are detailed in Section 2.2 below. The network, and the details of the data-processing pipeline generating the patches, are detailed in Section 2.3.

After upstream training on the initial dataset, the upstream network weights can be used to generate representations. By truncating the network, removing some of the last layers, we can extract an embedding of any input vector. These representations can be used in a *downstream* task to train a new network, with its own weights, minimizing a loss (typically classification or regression) between its output and the downstream task labels (see Figures 1 and 2).

We mostly evaluate our methods on downstream performance: the accuracy after training of the downstream network, on a task that is unknown during upstream training. However, the pretext reordering task can be of interest in and of itself, as in learning-to-rank problems (Liu, 2011), and we also report generalization performance in this task.

As an aside, in the Jigsaw puzzle reassembly task, Noroozi & Favaro (2016) show that the choice of permutation set matters when it comes to performance. They make use of the Hamming distance to

choose a permutation set that it maximally separated. This permutation set is diverse but is not close to covering the entire permutation space. By supporting the full set of permutations, our approach does not suffer from this issue.

The downstream tasks are dataset-dependent. However, the network in the upstream reordering task requires minimal modification across tasks and modalities (see Section 3 for details). In this work, we demonstrate the effectiveness of our method on audio, video, and images, across several classification and regression tasks.

## 2.2 DIFFERENTIABLE RANKING METHODOLOGY

Our methodology for representation learning relies importantly on the ability to incorporate ordering or ranking operations in an end-to-end differentiable pipeline. During training for the upstream task, the last two layers of the network consist of: a vector of score values $\theta_w(X) \in \mathbb{R}^n$, and network outputs $f_w(X) = y_\varepsilon^*(\theta_w(X)) \in \mathbb{R}^n$, using a differentiable ranking operator $y_\varepsilon^*$ that we detail here. The goal of the upstream task is to find good parameters $w$ such that the network outputs $f_w(X)$ correctly recover the label $y$ representing the permutation applied to the patches in $X$.

In earlier works using permutations as pretext task (Noroozi & Favaro, 2016; Lee et al., 2017), training with permutation labels is achieved by reducing the permutation inversion task to classification. More specifically, it is encoding a fixed number of permutations $L \ll n!$ as classes. Each class is represented by a one-hot vector, and network outputs are logits $\theta_w(X)$ of size $L$, leading to a prediction by taking a softmax among these $L$ classes. This approach is obviously limited: representing all the permutations requires in principle $n!$ classes, which is quickly not manageable, even for small values of $n$. Further, this does not take into account the similarity between permutations: with this encoding, permutations are orthogonal, no matter how similar they are.

We address these issues by having network outputs $\theta_w(X)$ of size only $n$, and interpreting their relative orders as the predicted permutation (e.g. $y = (0, 1, 2, 3)$ if they are in decreasing order, predicting the identity permutation). The pretext labels also encode the permutations in this manner. This gives a unique encoding to each permutation, operates in dimension $n$, and can be computed with sorting operations in time $O(n \log n)$. Further, small distances in these encodings naturally represent similar permutations.

However, applying directly a ranking operator $y^*$ on the last layer of the network would not allow for backpropagation of gradients: the function of the weights $w \mapsto L(y^*(\theta_w(X)); y)$ is piece-wise constant. Small changes in the weights $w$ induce either large jumps or no change in value at all, its gradients are 0 almost everywhere, and undefined otherwise. In order to overcome this matter, we consider instead two differentiable ranking operations, one using stochastic perturbations, introduced in Berthet et al. (2020) and another one using regularization (Blondel et al., 2020b). These operations, denoted here by $y_\varepsilon^*$, map any vector of $k$ values to a point in the convex hull of permutation encodings in dimension $k$ (e.g. $(0.1, 0.9, 2.2, 2.8)$ over 4 elements). They can be thought of analogous to the softmax operator for ranking. They share some of its properties: good approximation of the original function, differentiability in the input values $\theta$ with non-zero derivatives everywhere, ease of computation, and tuning by a temperature parameter $\varepsilon$ (see Appendix A.1 for further details).

Adjusting the network parameters $w$ requires a notion of loss function between $y_\varepsilon^*(\theta)$ and $y$. For the version of $y_\varepsilon^*(\theta)$ based on stochastic perturbations (Berthet et al., 2020), we use the associated *Fenchel–Young loss* (Blondel et al., 2020a), that act directly on $\theta = \theta_w(X)$ (outputs of the vector, and inputs of the sorting operations), written here as $L_{\mathsf{FY}}(\theta; y)$ (see Appendix A.1). This loss is convex in $\theta$, smooth, equal to 0 if and only if $y_\varepsilon^*(\theta) = y$. Its gradients are given by

$$\nabla_\theta L_{\mathsf{FY}}(\theta; y) = y_\varepsilon^*(\theta) - y \,.$$

We call this loss "Perturbed F-Y" in our empirical results. For the regularized version of $y_\varepsilon^*(\theta)$ (Blondel et al., 2020b), we use

$$\frac{1}{2}\|y_\varepsilon^*(\theta) - y\|^2.$$

We call this loss "Fast Soft Ranking (FSR)" in our empirical results. We opt for these two losses for their good theoretical properties and $O(n \log n)$ complexity. Other choices (Mena et al., 2018; Cuturi et al., 2019; Vlastelica et al., 2019; Rolínek et al., 2020; Grover et al., 2019) are also possible, potentially with higher computational cost, or regions with zero gradient.

## 2.3 Implementation and architecture

**Data-processing** When constructing the self-supervised task, inputs that can be treated as images (i.e. with width and height) are sliced into patches. This slicing is controlled by two variables $n_x$ and $n_y$, determining respectively the number of columns and rows used. In Noroozi & Favaro (2016), 9 square patches are used. Across modalities, different slicing scheme can lead to better performance. In video analysis, there is only one axis along which to permute: time. In audio processing, the downstream task may benefit from the pretext task using frequency slices of the spectrogram instead of time slices (see Figure 1 for illustration). We explore these questions Section 3.

**Upstream task.** For the reordering pretext task, performed upstream, we use a Context Free Network (CFN) from Noroozi & Favaro (2016). This network uses an AlexNet (Krizhevsky et al., 2012) backbone which processes each patch individually while sharing weights across all patches as shown in Figure 1. By processing each patch independently, but with shared weights, the network cannot rely on global structure. After the shared backbone, the patches are passed together through two fully connected layers. The layer represents the predicted ranks of the input permutation. Although CFN was originally designed for images, one of our empirical contributions is to show its application to a number of separate modalities without any change to the training mechanism (Figure 1). In particular, we show important performance gains on audio tasks.

**Downstream task.** For the downstream task, we use classifiers with low capacity: a 3-layer multi-layer perceptron (MLP) for audio and video task, and a linear classifier for images. It is trained on embeddings extracted at the first aggregate fully connected layer of the pretext network (whose weights are frozen during this part). The MLP's output layer is task dependent. For a regression downstream task, the output of the MLP is a single scalar and the downstream model is trained to minimize mean-squared error. On the other hand, for a downstream classification task, the output of the MLP is a softmax over the class logits, and we train the downstream model by minimizing the cross-entropy between the predictions and the true labels.

## 3 Experiments

In this section, we demonstrate the effectiveness of permutation-based self-supervised learning as measured by the test accuracy in downstream tasks across several modalities. All experiments were carried out in Tensorflow (Abadi et al., 2016) and run on a single P100 GPU. For all modalities, in addition to the downstream task, we also report the performance on the upstream task using partial ranks, the proportion of patches ranked in the correct position. We will open-source our codebase for reproducibility and reusability.

### 3.1 Audio

**Experimental setup.** The NSynth dataset (Engel et al., 2017) offers about 300,000 audio samples of musical notes, each with a unique pitch, timbre, and envelope recorded from 1,006 different instruments. The recordings, sampled at 16kHz, are 4 seconds long and can be used for 3 downstream classification tasks: predicting the instrument itself (1,006 classes) the instrument family (11 classes) and predicting the pitch of the note (128 classes). Since the pitch estimation is a regression task, we report in our results the mean squared error (MSE) for this task.

Audio processing systems usually take as input a log-compressed spectrogram of the recordings rather than the raw waveform, since it is a convenient and compact time-frequency representation of the input signal. Moreover, the 2D structure of the spectrogram allows us to use the 2D convolutional network as for images and videos.

We train our CFN with an AlexNet backbone on the upstream task of predicting the applied permutation for 1000 epochs, over mini batches of size 32 and with an Adam optimizer (Kingma & Ba, 2014) with a learning rate of $10^{-6}$. After convergence, we evaluate the downstream generalization performance over the 3 NSynth tasks, by replacing the last layers of the network by a 3-layer MLP and replacing the random permutation by the identity. To understand the quality of the produced embeddings, we vary the number of examples used to train train the downstream task and report the results for different data regimes.

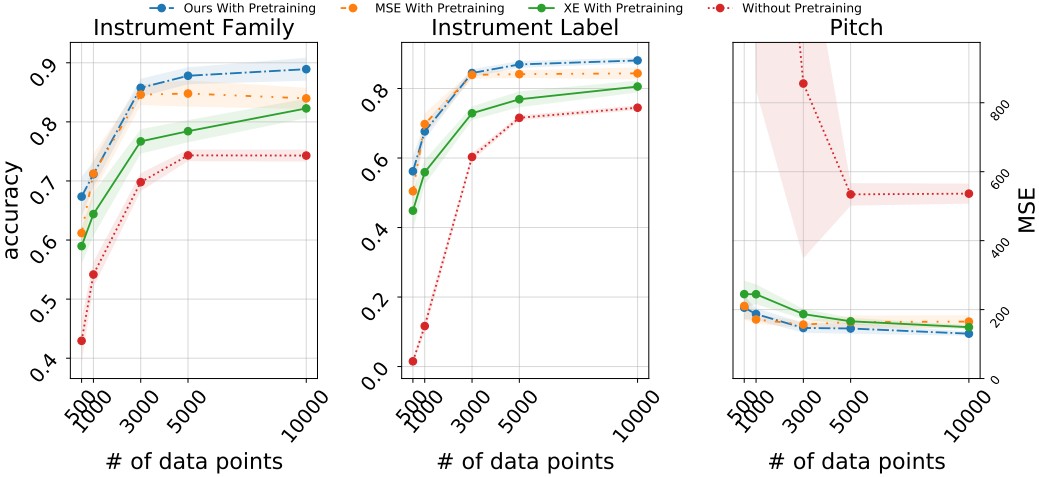

Figure 3: Performance of our permutation-based pretraining over 3 audio tasks when varying the number of data points in the downstream task.

| Task | Random Embedding | Fixed Permutation | Fast Soft Ranking | Perturbed F-Y | MSE |
|---|---|---|---|---|---|
| Instr. Family (ACC) | $0.46_{\pm 0.01}$ | $0.75_{\pm 0.02}$ | $0.84_{\pm 0.02}$ | $\mathbf{0.85}_{\pm 0.03}$ | $0.79_{\pm 0.02}$ |
| Instr. Label (ACC) | $0.35_{\pm 0.04}$ | $0.70_{\pm 0.03}$ | $0.72_{\pm 0.03}$ | $\mathbf{0.76}_{\pm 0.04}$ | $0.71_{\pm 0.03}$ |
| Pitch (MSE) | $258.76_{\pm 3}$ | $144.48_{\pm 18}$ | $133.85_{\pm 19}$ | $\mathbf{124.6}_{\pm 14}$ | $156.08_{\pm 21}$ |
| Partial Ranks Accuracy | - | - | $0.57_{\pm 0.0}$ | $\mathbf{0.58}_{\pm 0.0}$ | $0.51_{\pm 0.0}$ |

Table 1: Performance on three downstream tasks with 1000 downstream data points taken from NSynth. ACC stands for accuracy, and MSE for mean squared error.

We compare the different variants of our method (number and nature of the patches) with 2 baseline alternatives: i) training the downstream architecture on an untrained encoder (referred to as Random Embedding) and ii) solving the same upstream task but using instead a finite set of 100 permutations as proposed by Noroozi & Favaro (2016) (Fixed Permutation).

Lastly, we compare different losses to train the permutation pretext task: a) cross entropy (XE) when learning over 100 permutations, b) MSE loss (MSE), c) soft ranking via perturbations (Perturbed F-Y) and d) soft ranking (Fast Soft Ranking).

**Empirical results.** First, we compare the different methods for different data regimes in the downstream task and report graphically the results in Figure 3, choosing the accuracy for the classification tasks and MSE for the regression one. We observe that in the low data regime our method strongly outperforms fully supervised model trained end-to-end rather than on a fixed embedding. Moreover, even though the margin closes when adding more data, we observe that the features obtained greatly assist in the downstream learning process. We also observed that pretraining is particularly impactful for pitch estimation which aligns with results found by Gfeller et al. (2020).

We report in Table 1 the results for 1000 downstream examples (see 5 and 6, in the Appendix A.2 for 500 and 5000). Those experiments were run using 10 frequency bands, which corresponds to 10! permutations. This number of potential classes discards the use of classification loss: from a very practical point of view, such a huge number of classes would create a weight matrix on the last dense layer that would not fit in memory.

We first observe that random embeddings performs poorly but does represent a good baseline to be compared to. Second, when training the encoder with the fixed set of permutations, we observe a significant increase in performance that confirms experimentally the results reported by (Noroozi & Favaro, 2016). In the first three rows of Table 1, we report the results of our methods in the downstream metrics. We first observe that even with a mean squared error loss, the performance

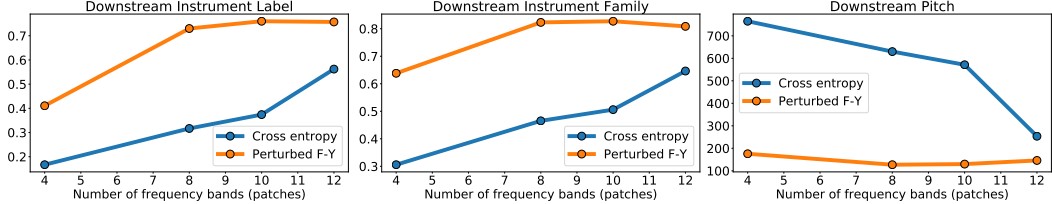

Figure 4: Performance on the NSynth dataset's downstream tasks, as a function of the number of frequency bands used for the pretext task on the audio experiment. With lower number of patches, the upstream task can become too easy to lead to good feature learning, risking overfitting.

| Task | Frequency | Time | Time-Frequency |
|---|---|---|---|
| Instr. Family (ACC) | $\textbf{0.82}_{\pm 0.003}$ | $0.79_{\pm 0.004}$ | $0.73_{\pm 0.003}$ |
| Instr. Label (ACC) | $\textbf{0.75}_{\pm 0.01}$ | $0.68_{\pm 0.002}$ | $0.45_{\pm 0.01}$ |
| Pitch (MSE) | $\textbf{136.68}_{\pm 12.63}$ | $206.11_{\pm 7.32}$ | $\textbf{137.81}_{\pm 13.02}$ |

Table 2: Slicing biases on Nsynth downstream tasks.

on the downstream task is comparable or better than the fixed permutation method and we show that using a ranking loss further increases the performance. Those results tend to confirm that i) permutation is an interesting pretext task, ii) considering all possible permutation helps building better representations and iii) the use of a ranking loss is the right choice of loss for such a task.

Furthermore, Fig.4 shows the effect of the number of frequency bands on the downstream performance. It shows that, as the number of permutations grows, the performance over the downstream task increases, providing better representations. However, it eventually stops increasing or can even decrease the performance.

We report in the last row of Table 1 performance in the pretext task. Good performance on the downstream task is often connected to good performance on the pretext task. Here, we measure performance by the ability of the CFN network to reorder the shuffled inputs, reporting the proportion of items ranked in the correct position.

**Time-frequency structure and permutations.** Unlike images, the horizontal and vertical dimensions of a spectrogram are semantically different, respectively representing time and frequency. While Noroozi & Favaro (2016) only exploited square patches, experimenting with audio allows exploring permutations over frequency bands (horizontal patches), time frames (vertical patches) or square time-frequency patches, and comparing the resulting downstream performance Table 2 reports a comparison between these three settings. Overall, shuffling along the frequency axis only is the best pre-training strategy. These results illustrate a particularity of the dataset: our inputs are single notes, many of them having an harmonic structure. In this context, learning the relation between frequency bands is meaningful both to recognize which instrument is playing, as well as which note (pitch). This also explains the poor performance of slicing along the time axis. Pitch is a time-independent characteristic, so the time structure is not relevant for this task. Moreover, musical notes have an easily identifiable time structure (fast attack and slow decay), which may make the task of reordering time frames trivial. We hypothesize that signals with a richer, non-stationary time structure, such as speech, would benefit more from shuffling time frames.

## 3.2 VIDEO

**Experimental setup.** Another modality where shuffling can assist learning is in video classification, as done by Lee et al. (2017). In this case, instead of slicing a single image with multiple patches, a time frame is treated as a patch and multiple frames are permuted. We sample these frames uniformly along the video, and use 20 frames. We experiment on the something-something dataset (Goyal et al., 2017), a dataset of videos labelled with 174 actions. We choose this dataset as many labels incorporate information about the dynamics (e.g. "Pulling something from right to left"), so that classification can be improved by learning the natural order of frames.

| Task | Fixed Permutation | Perturbed F-Y | Fast Soft Ranking | MSE |
|---|---|---|---|---|
| Label (ACC) | 0.13 ±0.01 | 0.19 ±0.001 | **0.21** ±0.002 | 0.12 ±0.03 |
| Partial Ranks | - | 0.27 ±0.003 | **0.29** ±0.001 | 0.17 ±0.03 |

Table 3: Up and downstream performance for video experiments.

| Task | Fixed | Perturbed F-Y | FSR | MSE |
|---|---|---|---|---|
| Label - 1000 (AUC Upper) | 0.81 ±0.04 | 0.84 ±0.02 | 0.84 ±0.03 | 0.84 ±0.04 |
| Label - 1000 (AUC Lower) | 0.80 ±0.03 | 0.82 ±0.02 | 0.82 ±0.04 | 0.82 ±0.05 |
| Label - 3000 (AUC Upper) | 0.83 ±0.03 | 0.85 ±0.02 | 0.85 ±0.02 | 0.85 ±0.02 |
| Label - 3000 (AUC Lower) | 0.82 ±0.02 | 0.84 ±0.02 | 0.84 ±0.02 | 0.84 ±0.02 |
| Partial Ranks | - | 0.14 ±0.02 | 0.14 ±0.02 | 0.14 ±0.03 |

Table 4: Upstream training on the ImageNet dataset with downstream task performance evaluated on Pascal VOC 2012 where performance is measured using Mean AUC.

**Experimental results.** We report in Table 3 results on self-supervised learning with permutations on frames of video. Competitive standards for fully supervised performance (using the whole video) on this dataset are around 0.65. Our method, with drastically reduced amount of training data (3000 videos, 1.4% of the dataset), operating on 20 frames, still reaches 0.21. Interestingly, when using fewer frames than 20, the downstream performance does not improve above random chance. Hence, scaling to more patches is required to in this setting, which makes using a fixed number of permutations challenging. This may explain the poorer performance of the Fixed Permutation baseline on this task, and points to permutations as a promising avenue of exploration for self-supervised learning on videos.

In addition to the potential for self-supervised learning, performance on permutation task itself is interesting in the context of videos. We also report experimental results on the upstream reordering task itself in Table 3. In this case, the permutations acts on 20 frames, with an order of $2.10^{18}$ possible permutations. Our results show that a significant fraction of elements are correctly reordered, even though this particular permutation has in all likelihood not been observed during training.

It is interesting to note that the methods that perform best on the pretext task also generally perform best on the downstream tasks. For example, we observe that our methods get higher performance than the MSE on the Jigsaw puzzle reassembly task. Correspondingly, our methods also achieve higher performance on the downstream tasks.

## 3.3 IMAGES

**Experimental setup.** We also applied our methodology to images, using the ImageNet and Pascal VOC 2012 datasets. In this experiment, we trained our CFN network on $128 \times 128$ images with $3 \times 3$ patches. We used a learning rate of $10^{-6}$, batch size of 32, and the Adam optimizer. We trained the same CFN network on the jigaw reassembly task using the ImageNet dataset. We then evaluated the quality of the embeddings via a downstream classification task on the PASCAL VOC 2012 dataset. We use the mean AUC metric to measure performance of our method which is appropriate due to the nature of the Pascal VOC classification task.

**Experimental results.** Consistent with the other results, we find performance of the two differentiable ranks methods to be consistently higher than learning reordering with classification over fixed permutations, however performance is consistent with MSE although the differentiable ranking methods have slightly lower variance. Additionally, we highlight the efficiency with which the upstream network learns the permutation inversion task. The results in Table 4 show the efficacy of our method on 1000 and 3000 downstream data points.

## 4 CONCLUSION

We present in this paper a general self-supervised learning method that uses permutations to learn high-quality representations. We demonstrate that our method outperforms previous permutation learning schemes by incorporating fully differentiable ranking as a pretext loss, enabling us to take advantage of all $n!$ permutations, instead of a small fixed set. In particular, we show significant improvements in low data regimes. We also demonstrate that our method can be applied seamlessly to improved downstream performance of several classification and regression tasks across audio, videos, and images.

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

## A    APPENDIX

### A.1    DIFFERENTIABLE RANKING AND FENCHEL-YOUNG LOSSES

The differentiable operators for ranking that we consider map a vector of values to the convex hull of ranking vectors - i.e. vector whose entries are 1 to $n$. This set $\mathcal{C}$ is referred to as the *permutahedron* (see, e.g., Blondel et al. (2020b)). Both our approaches rely on the fact that for a vector $\theta \in \mathbb{R}^n$, the vector of ranks of $\theta$ in ascending order is given by

$$y^*(\theta) = \arg\max_{y \in \mathcal{C}} \langle y, \theta \rangle \,.$$

The two variations on differentiable ranking that we introduce are based on modifications of this formulation. The approach in Berthet et al. (2020) is to introduce random perturbations $Z$ tuned by $\varepsilon > 0$ to the vector of values

$$y_\varepsilon^*(\theta) = \mathbb{E}[\arg\max_{y \in \mathcal{C}} \langle y, \theta + \varepsilon Z \rangle] \,.$$

Its value as well as its derivatives can be approximated by Monte-Carlo (MC) schemes (see Berthet et al., 2020, for details). In this work, we use a normal distribution for $Z$, and in learning experiments use $\varepsilon = 0.1$ and average over 10 MC samples. If we denote by $F_\varepsilon(\theta)$ the expectation of the maximum (rather than the argmax), and $\varepsilon\Omega$ its Fenchel dual over $\mathbb{R}^n$, we also have that

$$y_\varepsilon^*(\theta) = \arg\max_{y \in \mathcal{C}} \{ \langle y, \theta \rangle - \varepsilon\Omega(y) \} \,.$$

This is the approach favored in Blondel et al. (2020b), where the function $\Omega$ is not defined as the dual of an expectation, but instead as an explicit convex regularizer over $\mathcal{C}$. In this paper, we use $\Omega(y) = \frac{1}{2}\|y\|^2$. The corresponding Fenchel-Young loss (Blondel et al., 2020a), whose definition extends to other problems than ranking, is defined for such convex regularized problems by

$$L_{\mathsf{FY}}(\theta; y) = F_\varepsilon(\theta) + \varepsilon\,\Omega(y) - \langle \theta, y \rangle \,,$$

where we recall that $F_\varepsilon = (\varepsilon\Omega)^*$ (Fenchel dual). As described in the main text, its gradient over the parameter $\theta$ is given by

$$\nabla_\theta L_{\mathsf{FY}}(\theta; y) = y_\varepsilon^*(\theta) - y \,.$$

Stochastic gradients can therefore be obtained by Monte-Carlo approximation, which we use to minimize this loss in this work.

### A.2    ADDITIONAL EXPERIMENTAL RESULTS

Appendix Tables for additional information on the audio task (see next page).

| Task | Random Embedding | Fixed Permutation | Fast Soft Ranking | Perturbed F-Y | MSE |
|---|---|---|---|---|---|
| Instr. Family (ACC) | 0.44 $_{\pm 0.04}$ | 0.64 $_{\pm 0.04}$ | **0.73** $_{\pm 0.05}$ | **0.71** $_{\pm 0.05}$ | 0.62 $_{\pm 0.05}$ |
| Instr. Label (ACC) | 0.23 $_{\pm 0.04}$ | 0.52 $_{\pm 0.06}$ | **0.56** $_{\pm 0.05}$ | **0.55** $_{\pm 0.05}$ | 0.47 $_{\pm 0.05}$ |
| Pitch (MSE) | 256.20 $_{\pm 38.75}$ | **188.37** $_{\pm 28.77}$ | **186.03** $_{\pm 25.49}$ | **183.96** $_{\pm 26.45}$ | 276.59 $_{\pm 53.46}$ |

Table 5: Performance on three downstream tasks on the NSynth dataset, with **500** downstream data points. ACC stands for accuracy, and MSE for mean squared error.

| Task | Random Embedding | Fixed Permutation | Fast Soft Ranking | Perturbed F-Y | MSE |
|---|---|---|---|---|---|
| Instr. Family (ACC) | 0.59 $_{\pm 0.02}$ | 0.91 $_{\pm 0.01}$ | **0.93** $_{\pm 0.01}$ | **0.94** $_{\pm 0.01}$ | 0.90 $_{\pm 0.05}$ |
| Instr. Label (ACC) | 0.49 $_{\pm 0.04}$ | 0.89 $_{\pm 0.01}$ | **0.91** $_{\pm 0.01}$ | **0.91** $_{\pm 0.01}$ | 0.89 $_{\pm 0.01}$ |
| Pitch (MSE) | 251.45 $_{\pm 5.36}$ | **83.71** $_{\pm 8.55}$ | 97.98 $_{\pm 8.36}$ | 94.98 $_{\pm 8.05}$ | 108.46 $_{\pm 12.28}$ |

Table 6: Performance on three downstream tasks on the NSynth dataset, with **5000** downstream data points. ACC stands for accuracy, and MSE for mean squared error.

