# OpenReview forum: "Shuffle to Learn: Self-supervised learning from permutations via differentiable ranking"
_ICLR.cc/2021/Conference — Reject_

### Official Review · AnonReviewer3 · 2020-10-24
**The idea is interesting but it  lacks experimental support.**

**Rating:** 4
**Confidence:** 4

**Review:**

Summary:

The paper proposes to use differentiable ranking techniques to predict correct permutations. It can be applied to audio, video and images. It is quite obvious from the experiments that using differentiable ranking is more powerful to learn representations compared to fixed permutation in various applications.

Strengths:

- The idea is interesting and general to apply to different modalities.
- It is well written and easy to follow.

Concerns:

- Lack of comparison with state-of-the-art methods.
- In downstream tasks, there are always labels available, it is common to test on unsupervised setting as well.
- In Section 3.3, the experiment is conducted on MNIST, it would be more convincing to show results on larger datasets, since it is quite common to apply it on ImageNet.

---

### Official Review · AnonReviewer4 · 2020-10-29
**Soft ranking like loss can make jigsaw like unsupervised learning framework more feasible.**

**Rating:** 4
**Confidence:** 2

**Review:**

The paper applied differentiable ranking operator on unsupervised learning framework that uses permutation based pretext task. They evaluated the proposed approach in audio, video, and image classification tasks. The results show that the proposed differentiable ranking operator is showing better performance than the pre assigned fixed permutation.

Overall, the contribution of the paper is limited in applying the soft ranking loss (or perturbed FY) on permutation based unsupervised learning framework. I think there can be two factors that can improve the performance of the proposed approach to the baseline approach (fixed permutation). The first one can be related to the soft ranking (or perturbed FY) loss itself (unlike the classification of permutation cases or regression) and the second one can be related to the use of the number of permutation cases. By using the the soft ranking type loss, they could tackle the latter factor by using any permutation cases in the training phase. However, to verify where the performance gain come from, the authors can make an another comparison that is based on classification type loss (classification on N! permutation cases). It may be hard to deal with extremely large number of class labels, but at least some comparison table can be added with small sub dataset to verify the distinction between the two factors.

---

### Official Review · AnonReviewer1 · 2020-10-29
**Missing comparisons to related methods**

**Rating:** 4
**Confidence:** 4

**Review:**

This paper presents a self-supervised learning task of shuffling input patches and demanding the network to learn to unshuffle. A related prior work, Noorozi and Favaro (2016) uses a fixed set of permutations to do this task for a given number of patches, and the current paper argues to expand this idea for the full set of permutations. To this end, the paper encodes a permutation as a number tuple, with the goal of the network to learn to produce the correct tuple that has the numbers in order. As the numbers are discrete and thus non-differentiable, the paper suggests differentiable soft-variants using stochastic perturbations and regularizations (Fenchel-Young loss). Experiments are provided on audio and video tasks and show promise over the method of Noorozi and Favaro (2016).

Pros
1. A relatively simple self-supervised task with simple regularizations.
2. Experiments show some promise.

Cons:
1. Since the work of Noorozi and Favaro (2016), there have been several similar approaches to self-supervised learning that are not covered by the paper, either in the related work or in the experimental comparisons. A very reated work is
[a] Visual Permutation Learning, Santa Cruz et al., TPAMI 2018
that attempts to tackle the very specific problem of fixed set of permutations used in Noorozi and Favaro (2016). They also propose an end-to-end neural network for solving the the permutation finding task. The paper should contrast against this work both technically and in experiments.

[2] The experiments are rather weak in my opinion. As noted above, there are several prior papers (see the papers that cite the above paper) that attempts the unshuffling task in various ways, while the paper only makes comparison to the 2016 baseline paper. More recent baselines should be included in the comparisons and the advantages of the specific approach needs to be convincingly established.

Overall, while the idea is interesting, the paper does not place its contributions within recent prior works.

---

### Decision · Program_Chairs · 2021-01-07
**Final Decision**

**Decision:**

Reject

**Comment:**

There is clear consensus on this submission. Reviewers cite a lack of comparison
with recent state-of-the-art methods and experiments on more realistic datasets.
Though the reviewers find aspects of the approach interesting, the decision is
to reject.